# Manufacture of Whey Protein Hydrolysates Using Plant Enzymes: Effect of Processing Conditions and Simulated Gastrointestinal Digestion on Angiotensin-I-Converting Enzyme (ACE) Inhibitory Activity

**DOI:** 10.3390/foods11162429

**Published:** 2022-08-12

**Authors:** Marie Peslerbes, Angélica Fellenberg, Julien Jardin, Amélie Deglaire, Rodrigo A. Ibáñez

**Affiliations:** 1Departamento de Ciencias Animales, Facultad de Agronomia e Ingenieria Forestal, Pontificia Universidad Catolica de Chile, Macul, Santiago 7820436, Chile; 2École Supérieure D’agricultures Angers Loire, 49000 Angers, France; 3STLO, INRAE, Institut Agro, 35042 Rennes, France; 4Center for Dairy Research, University of Wisconsin-Madison, Madison, WI 53706, USA

**Keywords:** milk proteins, whey protein hydrolysate, bioactive peptides, papain, bromelain, ficin, hypertension, simulated gastrointestinal digestion

## Abstract

Hydrolysis of proteins leads to the release of bioactive peptides with positive impact on human health. Peptides exhibiting antihypertensive properties (i.e., inhibition of angiotensin-I-converting enzyme) are commonly found in whey protein hydrolysates made with enzymes of animal, plant or microbial origin. However, bioactive properties can be influenced by processing conditions and gastrointestinal digestion. In this study, we evaluated the impact of three plant enzymes (papain, bromelain and ficin) in the manufacture of whey protein hydrolysates with varying level of pH, enzyme-to-substrate ratio and time of hydrolysis, based on a central composite design, to determine the degree of hydrolysis and antihypertensive properties. Hydrolysates made on laboratory scales showed great variation in the type of enzyme used, their concentrations and the pH level of hydrolysis. However, low degrees of hydrolysis in papain and bromelain treatments were associated with increased antihypertensive properties, when compared to ficin. Simulated gastrointestinal digestion performed for selected hydrolysates showed an increase in antihypertensive properties of hydrolysates made with papain and bromelain, which was probably caused by further release of peptides. Several peptides with reported antihypertensive properties were found in all treatments. These results suggest plant enzymes used in this study can be suitable candidates to develop ingredients with bioactive properties.

## 1. Introduction

Short-chain peptides derived from food products have received special interest from scientists, manufacturers and consumers, since they exhibit certain beneficial biological activities and share similar structural sequences with endogenous biological peptides with specific physiological functions, such as hormones, neurotransmitters or those with regulatory responses, leading to agonistic or antagonistic activities [1]. These products, known as bioactive peptides, can also be released during fermentation and/or enzymatic processing of dairy products and ingredients due to hydrolysis of proteins [2]. Bioactive peptides derived from caseins (CN) and whey proteins (WP) have shown several biological functions, including anti-hypertensive, opioid, immune modulatory, antioxidant, antimicrobial and others [2,3,4,5]. Among the most studied bioactive peptides in human physiology are those associated with regulating the cardiovascular system [3,5,6], since arterial hypertension is the leading cause of death of more than 10 million people annually [7]. In the human organism, one of the main functions of the renin-angiotensin system (RAS) is associated with the regulation of blood pressure, in which the angiotensin-converting enzyme (ACE) leads to an increase of blood pressure through the synthesis of angiotensin II (a potent vasoconstrictor) from angiotensin I [3]. Inhibition of ACE (ACE-I) is considered as one of the main strategies to reduce hypertension [2]. Some C-terminal sequences of bioactive peptides may bind to ACE and hence reduce its activity in the organism [8]. Bioactive peptides derived from various dairy sources have shown efficacy in reducing hypertension by inhibiting ACE in both in vitro and in vivo studies [3,4,5,6,9,10].

Examples of commercially available products containing bioactive peptides with ACE-I are cheese [11], yogurt and fermented milks [12], as well as CN [13] and WP hydrolysates (WPH) [4,5]. The latter is an ingredient rich in bioactive peptides with several applications in the food industry [14] and can be manufactured from WP fractions obtained directly from cheese manufacture (i.e., sweet whey) or partially purified fractions obtained by further processing (i.e., whey protein concentrates, WPC; whey protein isolates, WPI) [4,14]. Sweet whey alone has also exhibited ACE-I activity, which depends on cheese manufacturing conditions and the extent of CN hydrolysis during cheesemaking caused by rennet and/or microbial enzymes [15]. Bioactive peptides found in WPH are derived from major WP (β-lactoglobulin, β-lg; α-lactalbumin, α-lac; bovine serum albumin and BSA), minor proteins (i.e., immunoglobulins, lactoferrin and lactoperoxidase) and CN fractions released during cheese manufacture (i.e., glycomacropeptide and others) [4,5,9]. WPH can be obtained using enzymes from animal, plant and microbial origins [2,3,4,5,9,10]. However, the use of animal enzymes could be limited by the food industry in coming years, due to cultural and religious concerns of consumers (i.e., vegetarians, Halal, Kosher, etc.) [16]; hence, the use of plant enzymes in the manufacture of WPH can be an alternative to explore their potential for improving ACE-I properties. Enzymes extracted from papaya (papain) [17,18,19,20], pineapple (bromelain) [18], Solanaceae flowers, berries, melon [21], Osage orange [22] and cardoon [23] have been previously been used to improve the ACE-I properties of WP or their fractions. These studies were performed at pH levels near the optimum properties of each enzyme. However, processing parameters such as pH, level of enzyme and time of hydrolysis can potentially modify the peptide composition of WPH [24], which can also influence their ACE-I properties. In addition, studies simulating enzymatic digestion processes, such as in vitro gastrointestinal digestion, are mandatory to identify the bioaccessibility of bioactive peptides in food systems, since this process may lead to the loss of functionality of certain biopeptides [1,3,8,17,25]. We hypothesize the ACE-I properties of WPH obtained with plant enzymes are greatly affected by pH, level of enzyme and time of processing. We also hypothesize that simulated gastrointestinal digestion (SGID) of WPH will further release ACE-I peptides and thus improve their bioactive properties. Hence, the objectives of this study were to assess hydrolysis of WP using three plant enzymes (papain, bromelain and ficin) at various levels of pH, enzymes and time of hydrolysis to yield increased ACE-I properties; and to determine the impact of SGID on the ACE-I properties and peptide bioaccessibility of selected treatments. This will potentially allow transforming whey-protein sources into value-added ingredients for the food industry.

## 2. Materials and Methods

### 2.1. Materials

Whey protein concentrate 80 powder [WPC80; 94.4% total solids, 77.8% total protein, 6.4% fat, 4.8% ash and <1% degree of hydrolysis (DH); Arla Foods Ingredients Group P/S, Viby, Denmark] was obtained from Granotec Chile S.A. (Santiago, Chile). Plant enzymes papain from papaya latex (P3375; 2.6 U/mg), bromelain from pineapple steam (B5144; 2.8 U/mg) and ficin from fig tree latex (F6008; 1.9 U/mg) were obtained from Millipore-Sigma (Merck KGaA, Darmstadt, Germany). The WPC80 powder was stored at room temperature and the enzymes at −20 °C. All other chemicals used for the study were purchased from Millipore-Sigma, Galenica S.A. and Winkler Ltda. (Santiago, Chile).

### 2.2. Preparation of Whey Protein Solutions and Enzymatic Hydrolysis on Bench-Scale

A stock solution of whey proteins containing 6.4% *w*/*w* total protein was prepared by dispersing WPC80 powder in deionized water for 4 h at 25 °C, followed by an overnight storage at 4 °C under continuous stirring to ensure complete solubilization. The whey protein stock was then subdivided to further dilute them to 4.0% *w*/*w* protein content prior to adjustments to various pH values (4.0–9.0) using 0.5 N HCl or 0.5 N NaOH, based on a central composite design (CCD; Table 1). Independent stock solutions of papain, bromelain and ficin (210 mg/mL) were diluted in 100 mM phosphate buffers with pH values adjusted to experimental conditions (Table 1) using 0.1 N HCl or 0.1 N NaOH.

According to the pH conditions described by the experimental design (Table 1), 160 mL of whey protein solutions were distributed in 250 mL bottles (Nalgene 2625F68, Thomas Scientific, Swedesboro, NJ, USA) and placed in a water bath (Model LSB-030S, LabTech, Jakarta, Indonesia) under continuous agitation at 150 rpm for 10 min to equilibrate the temperature to the specific treatment conditions (i.e., papain, 60 °C; bromelain and ficin, 50 °C; Table 1). After equilibration time, stock solutions of experimental enzymes were individually added to each treatment at varying concentration levels [enzyme-to-substrate ratios (E/S) of 0.14–0.42% *w*/*w* or ~1:22.5–~1:7.5 when expressed as enzyme-to-protein ratios; Table 1]. Since varying levels of enzyme were added to whey protein treatments, 100 mM phosphate buffers at the experimental pH and temperature values (Table 1) were used (if needed) to further dilute samples to target a protein content of 3.2% *w*/*w*. After time of hydrolysis under continuous agitation (10–490 min; Table 1), samples were heated at 95 °C × 15 min to stop enzymatic reactions and cooled in an ice bath for 30 min. Samples were adjusted to pH 7 at 20 °C using 0.5 N HCl or 0.5 N NaOH and centrifuged at 3000× *g* for 30 min at 20 °C to separate the insoluble fractions. The obtained supernatants (i.e., whey protein hydrolysates; WPH) were divided in two parts. One part was directly freeze dried, whereas the second part was fractionated by ultrafiltration using an Amicon^®^ stirred cell unit (Merck KGaA, Darmstadt, Germany) with a 3 kDa nominal molecular weight limit (NMWL) membrane filter as described by Lu et al. [11], followed by freeze drying of the UF permeates and retentates. The total and the UF WPH fractions were then stored at −20 °C for further analyses.

### 2.3. Compositional Analyses

The composition of the WPC80 used for this study was analyzed using AOAC official methods [26]. Levels of total solids was analyzed using the drying-oven method by heating the samples at 105 °C × 19 h, protein (%N × 6.38) by the Kjeldahl method, fat by Mojonnier method and ash by the gravimetric method heating sample at 550 °C × 4 h. In addition, the protein concentration of total and <3 kDa fractions from WPH were measured by the Kjeldahl method. All analyses were performed in duplicate.

### 2.4. Degree of Hydrolysis of Whey Protein Hydrolysates

The DH of WPH samples was estimated by the trinitrobenzenesulphonic acid (TNBS) method [27] according to the protocol described by Spellman et al. [28], which is based on the release of free amino groups using L-leucine as the reference standard. All analyses were performed in duplicate.

### 2.5. Inhibition of the Angiotensin-I-Converting Enzyme (ACE-I) of Whey Protein Hydrolysates

The in vitro ACE-I activity of the WPH samples was estimated using the spectrophotometric method described by Cushman and Cheung [29], following the protocol described by Lu et al. [11]. Before testing, total and <3 kDa fractions of WPH samples were diluted with deionized water at total protein levels ranging between 1–5 mg/mL and 0.1–0.8 mg/mL, respectively. Results were expressed as IC_50_ values, which are defined as the concentrations required to inhibit ACE to 50% of its original activity. All analyses were performed in duplicate.

### 2.6. In Vitro Static Digestion of Whey Protein Hydrolysates

Selected WPH samples were subjected to SGID, based on the INFOGEST standardized protocol [30,31]. Briefly, total WPH fractions were diluted to 40 mg/mL total protein content. Five mL of WPH solutions were placed in 50 mL Falcon centrifuge tubes (Corning Inc., Tewksbury, MA, USA) and tempered to 37 °C. The whole process was carried out in a water bath at 37 °C at constant agitation (150 rpm). The oral phase consisted of mixing samples with 5 mL of simulated salivary fluid (75 U/mL α-amylase, pH 7.0) and incubated for 2 min. In the gastric phase, 10 mL of simulated gastric fluid (2000 U/mL pepsin and 60 U/mL gastric lipase, pH 3.0) was added and incubated for 2 h. In the intestinal phase, 20 mL of simulated intestinal fluid (100 U/mL trypsin in pancreatin, 10mM bile salts, pH 7.0) and incubated for 2 h. The reaction was stopped by adding 80 μL of a proteinase inhibitor (0.5 M Pefabloc^®^ SC, Roche, Santiago, Chile), frozen at −80 °C and freeze dried. From each sample, half of the digested freeze-dried powder was then reconstituted in deionized water at its original concentration and subjected to ultrafiltration as previously described (Section 2.2). The digested-WPH <3 kDa fractions were freeze dried again and stored at −80 °C for further analyses. All analyses were performed in duplicate.

### 2.7. Peptide Analysis and Identification

Peptides from selected WPH samples with/without SGID were evaluated by mass spectrometry (MS) analysis according to the method described by Giribaldi et al. [32] with some modifications using a nano-rapid separation liquid chromatography (nano-RSLC) Dionex UltiMate^TM^ 3000 instrument equipped with a Q Exactive Orbitrap mass spectrometer and a nano-electrospray ion source (Thermo Fisher Scientific, San Jose, CA, USA). Freeze dried WPH samples were diluted to 40 μg/mL using an injection buffer and filtered through a 0.45 mm-filter. Five μL of sample was injected and concentrated in an Acclaim^TM^ PepMap 100m C18 guard column [300 μm internal diameter (i.d.) × 5 mm length, 5 μm particle size, 100 Å pore size; Dionex], followed by separation in an Acclaim^TM^ PepMap100 RSLC C18 column (75 μm i.d. × 250 mm length, 3 μm particle size, 100 Å; Dionex), using a mobile phase consisting of A [20 mL/L acetonitrile, 8 mL/L formic acid, 1 mL/L trifluoracetic acid (TFA) and 971 mL/L ultrapure water] and B (950 mL/L acetonitrile, 8 mL/L formic acid, 1 mL TFA and 41 mL/L ultrapure water). The elution gradient was at a flow rate of 0.3 μL/min, in which levels of solvent B were constant from 0 to 5 min, followed by a linear increase from 5 to 35% at 5–40 min and a second linear increase from 35 to 85% at 40–42 min to remain constant thereafter until 45 min; at 46 min levels of solvent B decreased to 3% and remained constant to 55 min, followed by an increase to 5% at 55.1 min and an equilibration time of 10 min for the next sample. Separated peptides were ionized with a Proxeon nanoelectrospray ion source. The capillary entering mass spectrometer was heated at 250 °C. The mass spectra were recorded in positive mode using the *m*/*z* range between 250–2000. The *m*/*z* resolution of the mass analyzer (200 atomic mass unit) was in acquisition mode to 70,000 and 17,500 for MS and MS/MS, respectively. The 10 most intense ions from each MS scan were selected for MS/MS fragmentation and ions with the same *m*/*z* were excluded from fragmentation for 15 s.

The software X!TandemPipeline [33] analyzed the obtained MS/MS spectra from tested samples to identify peptides using the *Bos Taurus* database from UniProt (http://uniprot.org, accessed on 8 July 2022; [34]). Any potential peptide from contaminant proteins were checked using the common Repository of Adventitious Proteins (cRAP; http://thegmp.org/crap, accessed on 8 July 2022) and removed from the dataset. Bioactive peptides from experimental samples were identified from the BIOPEP-UWM^TM^ database [35] accessed on October 2021. Validation of each identified peptide (e–value < 0.01) was performed to avoid occurrence of false positive results. The peptide abundance was determined based on the area under the curve from the extracted ion chromatograms (XIC) using MassChroQ software [36]. When a peptide was measured with several charge states, all ion intensities were summed.

### 2.8. Experimental Design and Statistical Analyses

The effects of pH, E/S and time of enzymatic hydrolysis of WPC80 using papain, bromelain and ficin were studied using a CCD and response surface methodology [37]. A 3-level factorial experimental design was used to investigate the effects of independent variables with 2-star points (α = 1.633) and 6 replicates of the center point (Table 1). Dependent variables were calculated using second-order polynomial models using stepwise regression to eliminate insignificant factors and thus simplify the model [37].

Comparisons between predicted and experimental responses were evaluated by one-sample Student’s *t*-tests. Selected experimental treatments were studied based on a 3 × 2 full factorial design using a general linear model to evaluate effects of enzyme (papain, bromelain and ficin), SGID (before or after) and their interactions on ACE-I properties. When significant differences were found (*p* < 0.05), means were analyzed by Tukey’s multiple comparison test.

The peptide abundances with ACE-I properties were processed for principal component analysis (PCA), following the procedure described by Piraino et al. [38]. All analyses were performed using Minitab^®^ Statistical software (Minitab Inc.^®^, State College, PA, USA).

## 3. Results

### 3.1. Degree of Hydrolysis and Inhibition of Angiotensin-I-Converting Enzyme from Whey Protein Hydrolysates Made with Papain, Bromelain and Ficin

The response surface plots of DH and ACE-I activity (expressed by IC_50_ values) in WPH made with papain are shown in Figure 1. A significant (R^2^ = 0.9495; *p* < 0.001) prediction model was obtained for DH from hydrolysates, in which pH and interactions of pH × E/S and pH × Time were negative terms, in contrast with Time, which was a positive term (Table 2). As expected, increasing time of hydrolysis from 30 to 480 min contributed to increased DH by a maximum of ~40% at low pH values (4.0) and high levels of E/S (Figure 1a,c,e,g). In contrast, an opposite trend was observed at increasing pH of hydrolysis, which contributed to a linear reduction of DH (<5% at pH 8.0). A significant (R^2^ = 0.7890; *p* < 0.01) prediction model was obtained from the IC_50_ values of the <3 kDa fraction of hydrolysates, in which pH was a negative term, in comparison with positive terms pH^2^, E/S^2^, Time^2^, pH × Time and E/S × Time interactions (Table 2). At 30 min of hydrolysis (Figure 1b), very high IC_50_ values (~320 μg/mL) were found at low pH values (~4.0) and low E/S levels, whereas an increase of time of hydrolysis up to 480 min contributed to reduced IC_50_ values under these conditions (i.e., low pH and low E/S levels; Figure 1d,f,h). However, increasing pH had a great impact on reducing IC_50_ values (<100 μg/mL at pH values of 8.0) at all times of hydrolysis.

The response surface plots of DH and ACE-I activity (expressed by IC_50_ values) in WPH made with bromelain are shown in Figure 2. A significant (R^2^ = 0.9065; *p* < 0.001) prediction model was obtained for DH from hydrolysates, in which E/S, Time and interaction E/S × Time were positive terms, in contrast with interactions pH^2^ and Time^2^, which were negative terms (Table 2). Time of hydrolysis contributed to an increase of DH (up to ~32%), especially under those conditions with increased levels of E/S, along with a little impact on reduced pH values (i.e., 4.0 vs. 9.0; Figure 2a,c,e,g). A significant (R^2^ = 0.6470; *p* < 0.001) prediction model was obtained from the IC_50_ values of the <3 kDa fraction of hydrolysates, in which E/S and Time were positive terms and interaction of Time^2^ was a negative term (Table 2). Despite pH having no impact, low IC_50_ values were found (≥30 μg/mL) at a combination of low E/S levels (<0.28%) and low time of hydrolysis (30 min; Figure 2b). Increasing levels of E/S led to high levels of IC_50_ values, which was greatly marked by an increase in time of hydrolysis up to 250 min (>170 μg/mL; Figure 2d,f). However, increasing hydrolysis time from 250 to 480 min showed no major impact on the response surface plot of IC_50_ values (Figure 2f,h).

The response surface plots of DH and ACE-I activity (expressed by IC_50_ values) in WPH made with ficin are shown in Figure 3. A significant (R^2^ = 0.8486; *p* < 0.001) prediction model was obtained for DH, in which pH, E/S and Time were positive terms, whereas the interaction Time^2^ was a negative term (Table 2). Increasing both E/S levels (from 0.14 to 0.42%) and pH values (from 4.0 to 9.0) led to increased DH values, which was more pronounced with increasing time of hydrolysis from 30 to 480 min, reaching DH values up to ~100% (Figure 3a,c,e,f). A significant (R^2^ = 0.4525; *p* < 0.05) prediction model was obtained from the IC_50_ values of the <3 kDa fraction of hydrolysates, in which pH and Time were positive terms and the interaction E/S^2^ was negative term (Table 2). A combination of low pH values along with low E/S levels led to decreased IC_50_ values, especially at reduced time of hydrolysis (i.e., 30 min; Figure 3b,d,f,h). However, the IC_50_ values obtained with ficin were considerably higher than the IC_50_ values found in papain and bromelain treatments (Figure 1 and Figure 2).

### 3.2. Predicted and Experimental In Vitro Antihypertensive Properties Obtained from Whey Protein Hydrolysates Made with Papain, Bromleain and Ficin under Selected Conditions

The strength of the ACE-I models obtained from WPH made with papain (Figure 1), bromelain (Figure 2) and ficin (Figure 3) was evaluated by selecting experimental conditions that yielded increased ACE-I (i.e., low IC_50_ values) within ranges of pH, E/S and time tested in models (Table 3). The predicted DH values obtained for papain, bromelain and ficin treatments were 6%, 6% and 40%, respectively. There were no significant differences between the predicted and experimental IC_50_ values found in papain (t = −0.42; *p* > 0.01) and bromelain (t = −0.52; *p* > 0.01) treatments; however, predicted and experimental results found in the ficin treatment differed (t = 11.03; *p* < 0.01). Hydrolysates made with papain and bromelain exhibited lower experimental IC_50_ values (i.e., <100 μg/mL) than those obtained with ficin, which also agrees with the findings in predictive models (Figure 1, Figure 2 and Figure 3).

### 3.3. Influence of Simulated Gastrointestinal Digestion on the In Vitro Anihypertensive Properties of Selected Whey Protein Hydrolysates Made with Papain, Bromelain and Ficin

The in vitro ACE-I properties obtained from selected WPH made with papain, bromelain and ficin (Table 3) before and after SGID are shown in Figure 4. As expected, whole hydrolysate fractions (Figure 4a) exhibited lower inhibition of ACE (IC_50_ values > 500 μg/mL) when compated to <3 kDa fractions (IC_50_ < 250 μg/mL; Figure 4b). As similarly found in the <3 kDa fractions (Table 3, Figure 4b), whole hydrolysates made from papain and bromelain showed lower IC_50_ values (i.e., higher in vitro antihypertensive properties) than the one made with ficin (Figure 4a; *p* < 0.05). The antihypertensive properties of whole hydrolysates were not affected by SGID (*p* > 0.05). In contrast, the <3 kDa hydrolysate fractions made with papain and bromelain exhibited an improvement in the ACE-I properties (i.e., a reduction of IC_50_ values), whereas the hydrolysate fraction from ficin showed a decrease in the antihypertensive properties (*p* < 0.05).

### 3.4. Identification of Antihypertensive Peptides in Selected Whey Protein Hydrolysates before and after Gastrointestinal Digestion

A summary list of ACE-I peptides identified in the <3 kDa fraction of selected WPH treatments (Table 3) before and after SGID is detailed in Table 4. A total of twelve different ACE-I peptides were found in selected WPH samples, including five ACE-I peptides from β-lg, three from α_s1_-CN, 2 from α_s2_-CN and two from β-CN, which contained from 6 to 16 amino acids. The type and number of peptides differed based on enzyme treatment (papain, bromelain and ficin), as well as the application of a SGID treatment (Table 4). WPH made from papain contained ten peptides with ACE-I activity, but only six were found after SGID, in which six peptides were degraded, whereas two peptides were released. Only five ACE-I peptides were found in WPH made with bromelain, but ten ACE-I peptides were found after SGID, in which one peptide was degraded and five peptides were released. WPH made from ficin contained ten peptides with ACE-I activity and five were found after SGID, in which six peptides were degraded and only one peptide was released. A biplot was obtained by PCA over the relative abundances of the <3 kDa ACE-I peptides from WPH samples before and after SGID (Figure 5). Two principal components (PC1 and PC2) accounted for 73% of the total variance. The PC1 separated samples among enzyme treatments before SGID (i.e., papain, bromelain and ficin), whereas PC2 separated samples by the SGID treatment, particularly for bromelain and ficin (Figure 5). Vector loadings showed that peptides derived from β-lg (LDAQSAPLR, DAQSAPLRVY and ALPMHIR) associated with papain enzyme treatment; peptides derived from α_s1_- and α_s2_-CN (RPKHPIKHQ and NMAINPSK, respectively) associated with bromelain enzyme treatment; peptides derived from β-lg (IPAVFK), α_s1_- (EIVPNSAEERLH), α_s2_- (FALPQYLK) and β-CN (LVYPFPGPIPNSLPQN and AVPYPQR) associated with ficin enzyme treatment; whereas peptides derived from β-lg (VLDTDYK) and α_s1_-CN (VPSERYL) associated with SGID for all treatments.

## 4. Discussion

The DH of WPH samples treated with papain was limited at high pH values, but exhibited an increase at low pH values (Figure 1a,c,e,g). In contrast, papain has been reported to be active among the pH range of 3–10, with an optimum activity at pH 6.5 [45]. According to Lieske and Konrad [46], hydrolysis of purified fractions of major WP, β-lg and α-lac, using papain is greatly increased by pH changes, due to structural changes of β-lg into a monomeric form and Ca^+2^ binding of α-lac that mostly occur at high (>7.5) and low (≤3.5) pH values, respectively. The same authors found that hydrolysis of commercial WPC can greatly differ in the extent of hydrolysis due to prior processing history of the WP material (i.e., heat treatment, extent of acid development during cheese manufacture, application of membrane filtration, etc.), as well as levels of E/S used for hydrolysis. WPH from papain has been previously made at pH values ranging from 5.5 and 7.0. As examples, Lieske and Konrad [47] found limited DH (<5%) in the hydrolysis of WP at E/S 1:50, pH 6.5 and 48 °C for 120 min; Ou et al. [48] observed > 20% DH when a WPC solution was hydrolyzed at an E/S of 3000 U/g protein, pH 5.5 and 54 °C for 300 min; Abadia-Garcia et al. [18] found ~12% of DH in sweet whey treated at a rate of 1:20 (E/S) pH 7 and 60 °C for 180 min; whereas Le Maux et al. [24] found reduced DH (~5%) when a WPC80 solution was hydrolyzed at a rate of 1:50 (E/S), pH 7 and 6.3 (with/without pH control) and 50 °C for 180 min. The DH found in WPH obtained with bromelain was considerably increased with higher E/S and time of hydrolysis, with little impact on pH (Figure 2a,c,e,g). Bromelain showed activity at pH between 4.0–9.0, with an optimum in the range 5.0–8.0 [49]. Only a few studies have evaluated the impact of bromelain on hydrolysis of WP, which has been tested at pH values in the range 6.5–7.5. Abadia-Garcia et al. [18] found ~9% of DH in WPH made with bromelain at a rate of 1:20 (E/S), pH 7 and 60 °C for 180 min; Ambrossi et al. [50] observed limited DH (~6%) in the hydrolysis of a 1% WPC solution using bromelain at an E/S 1:10 for 30 min at 45 °C and pH 6.5 and similar results (DH ~ 6%) were also obtained by the same research group in a recent study [51], in which hydrolysis was performed during 15 min at 50 °C at an E/S of 1:10 and pH 7.5; Li et al. [52] found 21% of DH in WP hydrolyzed with bromelain at E/S 1:10 for 90 min at 50 °C and pH 7; whereas Du et al. [53] observed ~12.5% DH in goat WP hydrolyzed at an E/S 9000 U/g protein for 360 min at 50 °C and pH 7. The DH of samples made with ficin considerably increased as levels of E/S, pH and time of hydrolysis increased (Figure 3a,c,e,g). According to Aider [54] ficin is active at pH values ranging among 5.0 and 8.0, although its optimum occurs at pH 7.5 [55]. In contrast with papain and bromelain, ficin exhibited a high proteolytic activity, even at short times of hydrolysis (Figure 1a, Figure 2a and Figure 3a). This observation is in accordance with the findings from Kheroufi et al. [55], in which ~20% DH was achieved using 0.5 and 1.0% E/S on WPC solutions hydrolyzed for 30 min at pH 7.5 and 80 °C.

The study of WPH has gained great interest due to the release of bioactive peptides with special focus on those exhibiting antihypertensive activity (i.e., ACE-I), in which enzymes of animal [9,19,25,39,40], microbial [25,56] and plant origin [17,18,19,20,21,22,23,57] were evaluated. The length of dairy-derived peptides exhibiting ACE-I activity usually comprises between 2–20 amino acids [2,3,6,9] and is generally associated with <3 kDa molecular size fractions as previously studied in WPH [57] and cheese [11]. Hence, the in vitro ACE-I activities reported in this study were focused on <3 kDa fractions. Based on our observations, improved ACE-I properties (i.e., low IC_50_ values) of <3 kDa fractions from WPH made with papain were associated with limited DH that occurred at high pH values of hydrolysis (DH < 10%; Figure 1). This could be explained by limited hydrolysis of WP leading to the release of peptides with biological activity, but further degradation of those peptides can lead to loss of their functionality [2]. In accordance with our findings, papain has been extensively used in the hydrolysis of WP to improve their ACE-I properties [18,19,20]. These hydrolysates have also shown other biological functions, including regulation of dipeptidyl peptidase IV [20,24] improvement in iron absorption [48], as well as antioxidant [18], antibacterial [58] and hypoglycemic activities [53], even when they are made with WP from non-bovine species (e.g., goat, camel, etc.). Similarly, improved ACE-I activity (i.e., reduced IC_50_ values) was observed in <3 kDa fractions of WPH from bromelain in those treatments with reduced DH (<16%) and time of hydrolysis, but only when an E/S < 0.28% was used. Our results are in accordance with the findings from Abadia-Garcia [18], who observed an increase of ACE-I properties of WPH made with bromelain. The use of bromelain has also shown antioxidant [18], antibacterial [59] and hypoglycemic activities [53] in WPH made using bovine or goat WP. Little work has been done on evaluating the impact of ficin on the hydrolysis of WP and this is the first study reporting ACE-I activity with ficin treatment (Figure 3a,d,f,h). However, these values were lower (i.e., higher IC_50_ values) than those obtained with papain and bromelain treatments. These differences could be attributed to the high proteolytic activity of ficin. A recent study has also shown antioxidant activity in WPH made from ficin [55].

Similarities between predicted and experimental ACE-I properties (expressed by IC_50_ values) obtained from selected WPH papain and bromelain treatments (Table 3) could be associated due to increased R^2^ values (≥0.60) from predictive models, in contrast with ficin in which a lower R^2^ value (~0.45) led to slight differences between predicted and experimental IC_50_ values. However, those differences were minimal (Table 3; ≤15 μg/mL). As previously stated, peptides with ACE-I activity have low molecular size (<3 kDa), hence whole WPH fractions obtained from papain, bromelain and ficin exhibited lower ACE-I activity (IC_50_ > 450 μg/mL; Figure 4a) than the UF fraction containing <3 kDa peptides (IC_50_ < 240 μg/mL; Figure 4b). As expected, these fractions contained peptides with ACE-I activity (Table 4), and this activity may be attributed to the presence of hydrophobic amino acids [i.e., alanine (A), isoleucine (I), leucine (L), methionine (M), phenylalanine (F), tryptophan (W), tyrosine (Y) and valine (V)] within the first three positions in the C-terminal region [23,25,60], as found in LDAQSAPLR, IPAVFK, VLDTDYK, ALPMHIR, EIVPNSAEERLH and FALPQYLK; the occurrence of proline (P), Y, F, W, L, A and V in the C-terminal region [6], as found in DASQAPLRVY and VPSERYL; as well as the occurrence of P near the C-terminal region [23], as in NMAINPSK, LVYPFPGPIPNSLPQN and AVPYPQR. Peptide RPKHPIKHQ also exhibited ACE-I activity, as reported in Table 4. However, the nano-RSLC-MS method used in the present study only allowed detection of peptides containing six or more amino acids [61], which probably limited identification of short peptides (two to five amino acids) with strong ACE-I properties, as reported in various WPH studies [2,9,17,20,23,25,39]. The occurrence of CN-derived peptides with ACE-I activity (Table 4) has been previously reported in WPH samples [23] and their precursors may be released during cheese manufacture, further drained with sweet whey (i.e., raw material for WPC manufacture) [15] and remain in WPC after processing. Major differences found in the extent of hydrolysis of WPC using papain, bromelain and ficin could be associated with the specificity of each enzyme. Hence, standardization of enzymatic activity in WP substrates is critical to achieve a DH that yields improved ACE-I properties. The latter can help processors to avoid excessive DH, as occurs with ficin, that lead to reduced ACE-I activity, but it may also contribute to the development of undesirable bitter taste, as Leksrisompong et al. [14] found a positive correlation between the DH of 22 commercial WPH (DH varying between 1 and 47%) and sensory bitterness, which was also associated with an increased occurrence of low molecular peptides (<1 kDa).

Despite SGID showing no effect on the ACE-I activity of whole WPH fractions (Figure 4a), the <3 kDa fractions obtained after SGID (Figure 4b) from treatments made with papain and bromelain exhibited a decrease in IC_50_ values, whereas in the ficin treatment the IC_50_ increased. These differences could be attributed to a combined interaction of degradation/release of bioactive peptides (Table 4), as also suggested by the PCA biplot obtained from relative abundance of ACE-I peptides (Figure 5). We also believe a low DH of WPH may lead to an increased ACE-I activity after SGID, as observed in our study in which papain and bromelain treatments (DH ~ 6%) had a decrease in IC_50_ values after SGID, whereas in the ficin treatment (DH ~ 40%) there was an increase thereafter (Figure 4b). This trend was also observed by Bustamante et al. [25] in WPH made with alcalase (DH ~ 37%) and flavourzyme (DH ~ 46%) that exhibited a decrease in ACE-I activity after they were treated with SGID, whereas a WPH made with chymotrypsin (DH ~ 19%) had no changes in ACE-I activity after SGID. Hence, the final impact of WPH on the ACE-I properties is not only impacted by their enzymatic treatment; it will also depend on SGID [17,25], as well as aminopeptidases from the intestinal brush border in biological systems [62]. Therefore, further studies in biological systems are necessary to validate these findings.

## 5. Conclusions

The results from this study suggest the use of plant enzymes are suitable to produce potential ingredients from WP with improved biological activity, such as that of ACE-I. WPH obtained from papain and bromelain yielded the highest ACE-I properties and exhibited reduced DH, in contrast with ficin treatments. Improvement in ACE-I properties of treatments was associated with the release of identified peptides with biological activity, as evaluated by nano-RSLC-MS. However, this technique did not allow the identification of bioactive peptides containing five or fewer amino acids. SGID performed to selected WPH treatments from papain and ficin led to an increase of ACE-I activity of the <3 kDa fraction obtained after digestion, which suggests further release of peptides with biological activity; in contrast with ficin treatment that exhibited a reduction of biological activity. Future studies will be required to evaluate the technical and economic feasibility of manufacturing WPH using these plant enzymes on an industrial scale, since food-grade papain, bromelain and ficin are commercially available in the market at competitive prices, and therefore it could provide an alternative approach to produce WPH with functional properties. The identification/quantification of short ACE-I peptides using supplemental chromatographic-MS techniques, their biological activity in in vivo studies and their technological functionality will also be addressed to completely characterize these ingredients.

## Figures and Tables

**Figure 1 foods-11-02429-f001:**
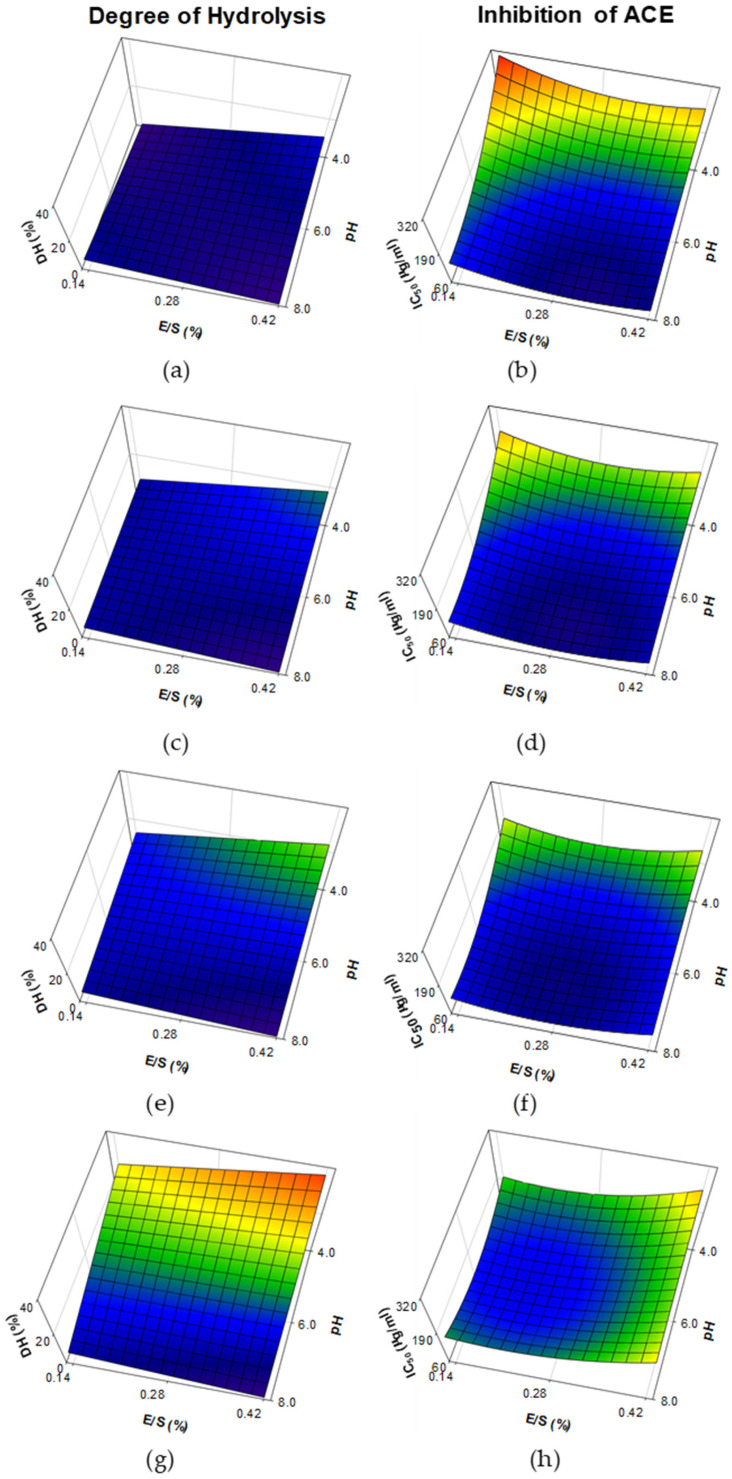
Response surface plots for the effect of pH and enzyme-to-substrate ratio (E/S) on the degree of hydrolysis and inhibition of angiotensin-I-converting enzyme (expressed as concentration of protein to inhibit ACE to 50%; IC_50_) in the <3 kDa fraction of whey protein hydrolysates made with papain. Prediction models were plotted at reaction times of 30 (**a**,**b**), 150 (**c**,**d**), 250 (**e**,**f**) and 480 min (**g**,**h**).

**Figure 2 foods-11-02429-f002:**
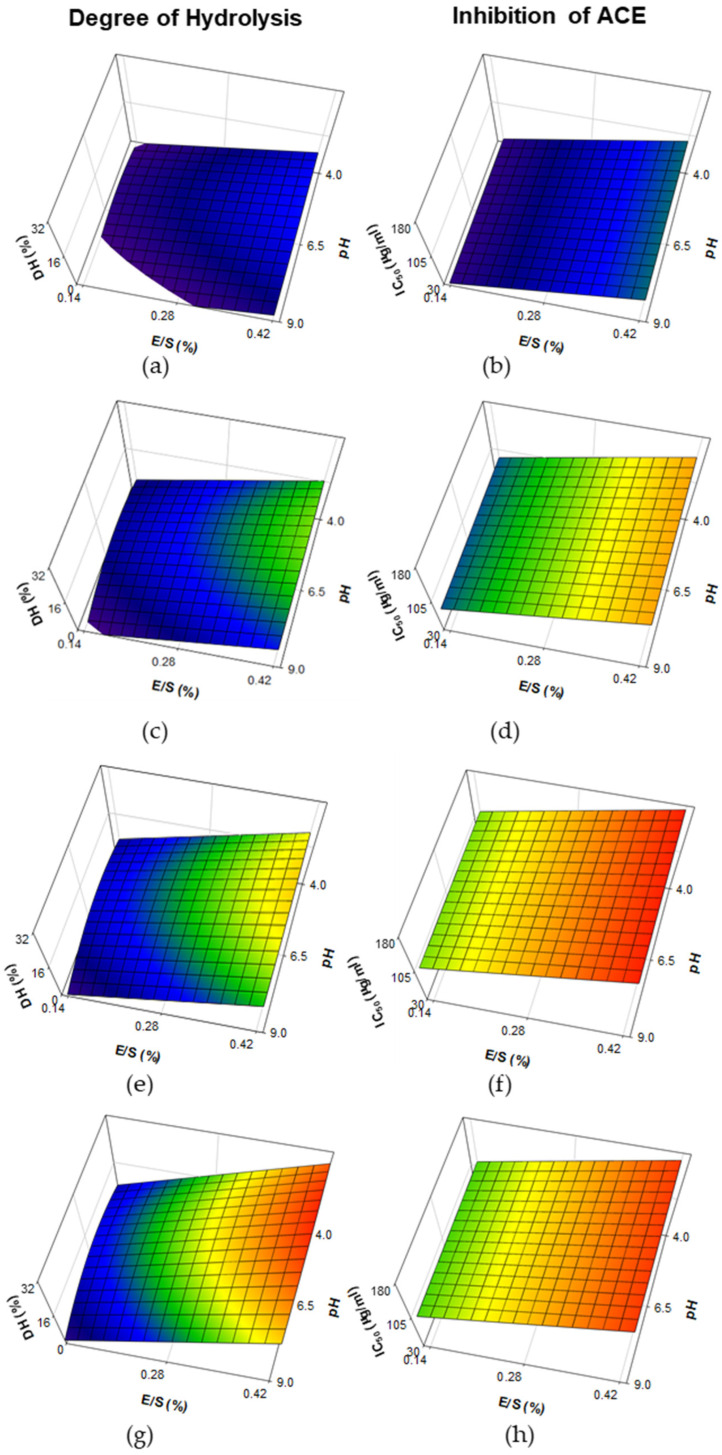
Response surface plots for the effect of pH and enzyme-to-substrate ratio (E/S) on the degree of hydrolysis and inhibition of angiotensin-I-converting enzyme (expressed as concentration of protein to inhibit ACE to 50%; IC_50_) in the <3 kDa fraction of whey proteins hydrolysates made with bromelain. Prediction models were plotted at reaction times of 30 (**a**,**b**), 150 (**c**,**d**), 250 (**e**,**f**) and 480 min (**g**,**h**).

**Figure 3 foods-11-02429-f003:**
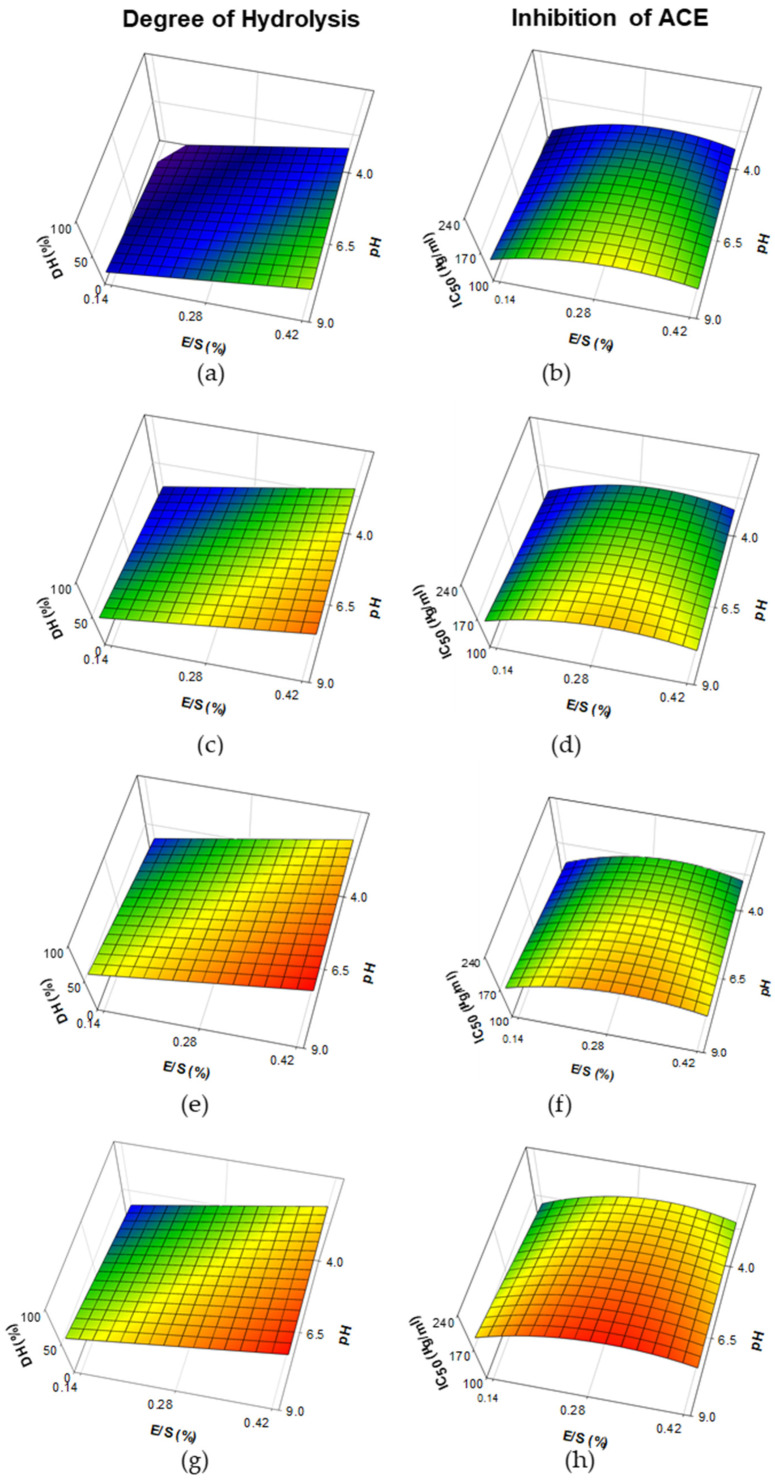
Response surface plots for the effect of pH and enzyme-to-substrate ratio (E/S) on the degree of hydrolysis and inhibition of angiotensin-I-converting enzyme (expressed as concentration of protein to inhibit ACE to 50%: IC_50_) in the <3 kDa fraction of whey proteins hydrolysates made with ficin. Prediction models were plotted at reaction times of 30 (**a**,**b**), 150 (**c**,**d**), 250 (**e**,**f**) and 480 min (**g**,**h**).

**Figure 4 foods-11-02429-f004:**
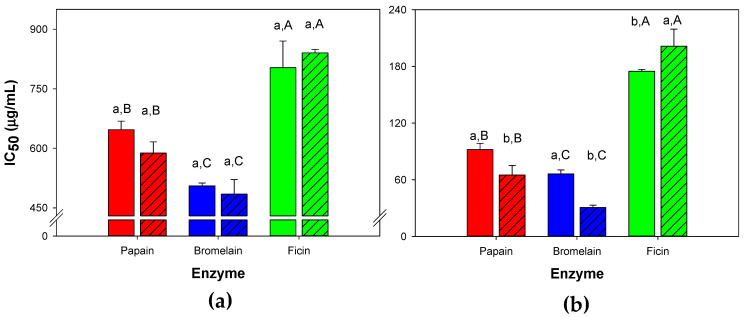
Inhibition of angiotensin-I-converting enzyme (ACE-I) obtained from the whole (**a**) and the <3 kDa fractions (**b**) of whey protein hydrolysates^1^ before (closed bars) and after simulated gastrointestinal digestion (closed bars with parallel lines). Values represent means and standard deviations (n = 3). ^1^ Selected whey protein hydrolysate treatments are detailed in Table 3. ^a,b,c^ Means within the same enzyme treatment not sharing a common superscript differ (*p* < 0.05). ^A,B,C^ Means among different enzyme treatments (with or without simulated gastrointestinal digestion) not sharing a common uppercase superscript differ (*p* < 0.05).

**Figure 5 foods-11-02429-f005:**
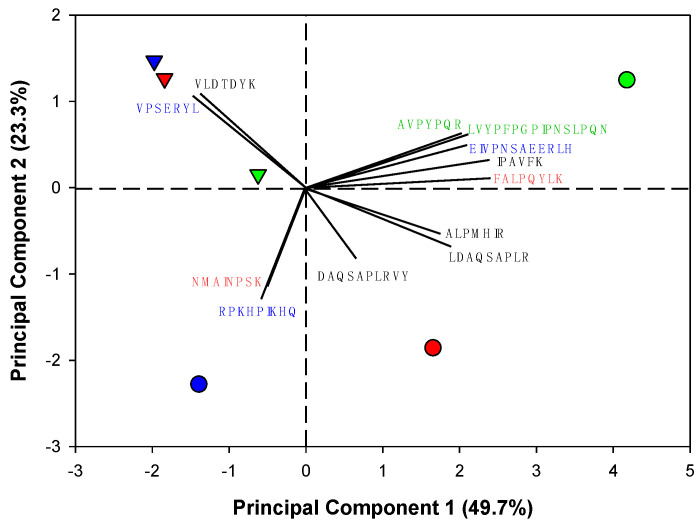
Biplot obtained by principal component analysis (PCA) from the abundances of the <3 kDa peptides with inhibitory angiotensin-I-converting enzyme (ACE) activity found in selected experimental whey protein hydrolysates made with papain (red symbols), bromelain (blue symbols) and ficin (green symbols), before (circle) and after (inverted triangle) simulated gastrointestinal digestion. Color of peptide sequences indicates parent proteins (β-lg, black; β_s1_-CN, blue; α_s2_-CN, red; α-CN green).

**Table 1 foods-11-02429-t001:** Experimental design indicating the coded and the actual values for pH, enzyme-to substrate ratio (E/S) and reaction time of hydrolysis of whey protein concentrate solutions.

Treatment Number	Coded Values			Actual Values			
	pH	E/S	Time	pH Papain	pH Bromelain and Ficin	E/S (%)	Time (min)
1	−1	−1	−1	4.78	5.00	0.194	103
2	+1	−1	+1	7.22	8.00	0.194	397
3	+1	+1	−1	7.22	8.00	0.366	103
4	−1	+1	+1	4.78	5.00	0.366	397
5	0	0	0	6.00	6.50	0.280	250
6	0	0	0	6.00	6.50	0.280	250
7	+1	−1	−1	7.22	8.00	0.194	103
8	−1	−1	+1	4.78	5.00	0.194	397
9	−1	+1	−1	4.78	5.00	0.366	103
10	1	+1	+1	7.22	8.00	0.366	397
11	0	0	0	6.00	6.50	0.280	250
12	0	0	0	6.00	6.50	0.280	250
13	−α ^1^	0	0	4.00	4.00	0.280	250
14	+α	0	0	8.00	9.00	0.280	250
15	0	0	−α	6.00	6.50	0.280	10
16	0	0	+α	6.00	6.50	0.280	490
17	0	−α	0	6.00	6.50	0.140	250
18	0	+α	0	6.00	6.50	0.420	250
19	0	0	0	6.00	6.50	0.280	250
20	0	0	0	6.00	6.50	0.280	250

^1^ α = 1.633.

**Table 2 foods-11-02429-t002:** Second-order polynomial models describing degree of hydrolysis (DH) and inhibition of angiotensin-I-converting enzyme (ACE-I) in the <3 kDa ultrafiltration fraction (expressed as the peptide concentration to inhibit ACE in 50%; IC_50_) obtained from the hydrolysis of whey protein concentrate using papain, bromelain and ficin at varying levels of enzyme-to-substrate (E/S) ratio, pH and reaction times.

Enzyme	Dependent Variable	Independent Variable	Coefficient	R^2^ (Adjusted) ^1^	*p*-Value
Papain	Degree of hydrolysis (DH)	Constant pH *** Time *** pH × E/S ** pH × Time ***	11.458 −6.645 7.028 −4.15 −6.86	0.9495	<0.001
	IC_50_ of <3 kDa fraction	Constant pH *** pH^2^ ** E/S^2^ * Time^2 +^ pH × Time * E/S × Time ^+^	111.36 −44.75 38.80 34.10 31.70 36.40 29.00	0.7890	0.001
Bromelain	Degree of hydrolysis (DH)	Constant E/S *** Time *** pH^2^ * Time^2^ ** E/S × Time ^+^	15.512 7.418 7.590 −2.86 −3.21 3.18	0.9065	<0.001
	IC_50_ of <3 kDa fraction	Constant E/S* Time *** Time^2^ **	143.51 28.60 46.30 −49.50	0.6470	<0.001
Ficin	Degree of hydrolysis (DH)	Constant pH ** E/S *** Time *** Time^2^ **	63.14 11.64 18.50 22.41 −23.21	0.8486	<0.001
	IC_50_ of <3 kDa fraction	Constant pH * Time * E/S^2 +^	187.20 16.71 22.64 −24.50	0.4525	0.013

^1^ R^2^ values were adjusted for the degree of freedom. ^+^
*p* < 0.1; * *p* < 0.05; ** *p* < 0.01; *** *p* < 0.001.

**Table 3 foods-11-02429-t003:** Comparison of angiotensin-I-converting enzyme (ACE) inhibitory activity (expressed as IC_50_ values) of the <3 kDa fraction of selected whey protein hydrolysates made with papain, bromelain and ficin predicted by the model with experimental results.

Enzyme	Treatments			Responses	
				IC_50_ (μg/mL)	
	pH	E/S (%)	Time (min)	Predicted ^1^	Experimental ^2^
Papain	7.5	0.28	150	93.6	91.9 ± 6.6
Bromelain	6.5	0.28	30	65.1	66.2 ± 4.2
Ficin	4.0	0.28	150	162.0	174.8 ± 2.0

^1^ 95% confidence interval; ^2^ Values represent mean and standard deviation (*n* = 3).

**Table 4 foods-11-02429-t004:** Angiotensin-I-converting enzyme (ACE)-inhibitory peptides ^1^ found in the <3 kDa fractions of whey protein hydrolysates ^2^ before and after simulated gastrointestinal digestion.

Peptide Sequence	ACE-Inhibitory Peptide	Papain		Bromelain		Ficin		Reported IC50 (mM)
		Before ^3^	After ^4^	Before ^3^	After ^4^	Before ^3^	After ^4^	
LDAQSAPLR	β-lg f(32–40)	+ ^5^	+	+	+	+	+	635.0 [9]
DAQSAPLRVY	β-lg f(33–42)	+	+	−	+	−	+	12.2 [17]
IPAVFK	β-lg f(79–83)	+	−	−	−	+	−	144.8 [39]
VLDTDYK	β-lg f(94–100)	−	+	−	+	+	−	946.0 [9]
ALPMHIR	β-lg f(142–148)	+	+	+	+	+	+	42.6 [40]
RPKHPIKHQ	α_s1_-CN f(1–9)	+	−	+	−	−	−	13.4 [41]
VPSERYL	α_s1_-CN f(86–92)	−	+	−	+	+	−	232.8 [42]
EIVPNSAEERLH	α_s1_-CN f(110–121)	+	−	−	−	+	+	NA ^6^ [43]
NMAINPSK	α_s2_-CN f(25–32)	+	+	+	+	+	−	60.0 [44]
FALPQYLK	α_s2_-CN f(174–181)	+	−	−	+	+	−	4.3 [44]
LVYPFPGPIPNSLPQN	β-CN f(11–26)	+	−	+	+	+	+	71.0 [12]
AVPYPQR	β-CN f(177–183)	+	−	−	+	+	−	274.0 [40]
Number of total ACE-inhibitory peptides		10	6	5	9	10	5	-

^1^ Peptides with ACE- I activity were identified by nano-RSLC-MS using the BIOPEP-UWM^TM^ database [35]. ^2^ Whey protein hydrolysates were made under the conditions detailed in Table 3. ^3^ From treatment before simulated gastrointestinal digestion. ^4^ From treatment after simulated gastrointestinal digestion. ^5^ “+” indicates that the peptide was present in the sample/treatment; “−” indicates that the peptide was not detected in the sample/treatment. ^6^ Information not available.

## Data Availability

Data is contained within the article and available at request from the corresponding author.

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
