# Peer review of "Manufacture of Whey Protein Hydrolysates Using Plant Enzymes: Effect of Processing Conditions and Simulated Gastrointestinal Digestion on Angiotensin-I-Converting Enzyme (ACE) Inhibitory Activity"

_foods, 2022, doi:10.3390/foods11162429_

Round 1

Reviewer 1 Report

I found the manuscript by Peslerbes et al., scientifically sound without evident flaws.

My main concern is about the importance of the findings are described and if they deserve publication on Foods.

In my opinion an original result is lacking, reporting the paper mainly the optimization of experimental conditions to find a bioactive hydrolyzed of WP using specific plant-derived enzymes. Probably a more specialized journal could be more suitable.

As minor note I would suggest the authors to improve the experimental methods adding the experimental condition they used instead of just referring to literature.

Author Response

I found the manuscript by Peslerbes et al., scientifically sound without evident flaws.

My main concern is about the importance of the findings are described and if they deserve publication on Foods. In my opinion an original result is lacking, reporting the paper mainly the optimization of experimental conditions to find a bioactive hydrolyzed of WP using specific plant-derived enzymes. Probably a more specialized journal could be more suitable.

Reply: The Authors appreciate the comment from the Reviewer. We are submitting this manuscript paper in a Special Issue “Research and Development of Functional Peptide in Foods”. We believe this manuscript would meet MDPI Foods requirements for publication in this Special Issue. If not, MDPI Foods also has a Dairy Section that would meet the same requirements as a specialized journal in the dairy processing field.

As minor note I would suggest the authors to improve the experimental methods adding the experimental condition they used instead of just referring to literature.

Reply: We included as detailed as possible all the methods used in this study on those experimental conditions that are not necessarily referenced on standard methods. On the other hand, several of the methods listed in the manuscript are standard methods that we believe are not necessary to include in such level of detail, as any reader can directly check the references for further information. Otherwise, it would lead to a manuscript with a extensive materials and methods section.

Reviewer 2 Report

The manuscript is very well written; clear, precise, and easy to understand.  Some comments should be addresed before their publication in Foods.

I suggest to the authors to compare statistically in the Table 3 the predicted vs experimental IC500 values.

Please specify what variables constitutes the PC1 and PC2. Figure 5.

Author Response

The manuscript is very well written; clear, precise, and easy to understand.  Some comments should be addresed before their publication in Foods.

I suggest to the authors to compare statistically in the Table 3 the predicted vs experimental IC500 values.

Reply: The authors appreciate the comment from the Reviewer. The predicted vs experimental IC50 values were evaluated by a t-test (as detailed in lines 225-230). These results are detailed in section 3.2, specifically in lines 312-317.

Please specify what variables constitutes the PC1 and PC2. Figure 5.

Reply: The authors appreciate the comment from the Reviewer. That information is detailed on the Results Section 3.4 of the manuscript, when describing results obtained from PCA Biplot (lines 371-373).

Reviewer 3 Report

The manuscript is interesting and it is well written. It is one of the uses that can be made of by-products from the food industry. I believe that the authors should emphasize the economic part of this study. Is it economically profitable to use the proposed enzymes? Are they easy to obtain? All this in order to emphasize the industrial use.

Other comments:

Line 109: Could pH modification an impact by themselves in the hydrolyzation of whey proteins?

Line 116: Why the pHs are different between papain and Bromelain and Ficin?

Line 352: Please do not duplicate information that is present in Table 4. As written this section maybe it is a bit difficult to follow with all the sequences of the peptides. Please rewrite.

Table 4: Please add a name to the second column where b-lg f(32-40)…..appear.

Author Response

The manuscript is interesting and it is well written. It is one of the uses that can be made of by-products from the food industry. I believe that the authors should emphasize the economic part of this study. Is it economically profitable to use the proposed enzymes? Are they easy to obtain? All this in order to emphasize the industrial use.

Reply: The Authors appreciate the comment from the Reviewer. Lines 523-531 of the manuscript were modified to include relevant information regarding on how addressing future studies are necessary to scale-up the manufacture of whey protein hydrolysates using commercially available papain, bromelain and ficin. These are commercialized by various enzyme houses (e.g., Enzyme Development Corporation, New York, NY). In fact, we are currently working with a whey processor in Chile to obtain WPH products on large-scale manufacture, using the enzymes evaluated in this study.

Other comments:

Line 109: Could pH modification an impact by themselves in the hydrolyzation of whey proteins?

Reply: Very low (pH < 2) or very high (pH > 10) pH values alone could cause damage to whey proteins that could eventually impact on hydrolysis. The pH values used in our study were based on information found in the literature, as well as preliminary work.

Line 116: Why the pHs are different between papain and Bromelain and Ficin?

Reply: Despite the literature described that papain could be active at pH values ranging from 3 to 10, we observed that when papain was evaluated at high pH values (pH > 8.0), there was very little or no impact on the hydrolysis of whey proteins (as observed in all predictive models found in Figure 1, where even at pH 8, the DH was minimal, or close to 0). This is the reason we decided the narrow the experimental conditions when tested papain.

Line 352: Please do not duplicate information that is present in Table 4. As written this section maybe it is a bit difficult to follow with all the sequences of the peptides. Please rewrite.

Reply: Duplicated information was removed from the manuscriot. Instead, results are now generally described our findings (Lines 360-369), that can be supplemented with information from Table 4.

Table 4: Please add a name to the second column where b-lg f(32-40)…..appear.                                                                                                                                       

Reply: As requested by the Reviewer, Change to Table 4 was made accordingly

Reviewer 4 Report

Well designed research paper with sound data that support the conclusions. The only correction suggested is to rephrase the L45.

Author Response

Well designed research paper with sound data that support the conclusions. The only correction suggested is to rephrase the L45.

Reply: The Authors appreciate the comment from the Reviewer. As requested, paragraph was re-phrased.

Reviewer 5 Report

The production of ACE-inhibitory peptides from whey proteins is already well  documented in the published literature, however the work reported in the present paper may be useful for applications using enzymes from vegetal origin.

Author Response

The production of ACE-inhibitory peptides from whey proteins is already well  documented in the published literature, however the work reported in the present paper may be useful for applications using enzymes from vegetal origin.

Reply: The Authors appreciate the comment from the Reviewer.